# Abundance and Phenotypic Diversity of the Medicinal *Sideritis Scardica* Griseb. in Relation to Floristic Composition of Its Habitat in Northern Greece

**Pinelopi K. Papaporfyriou, Eirini Sarrou, Eleni Avramidou and Eleni M. Abraham ***

Laboratory of Range Science (P.O.Box 236), Department of Forestry and Natural Environment, Aristotle University of Thessaloniki, 541 24 Thessaloniki, Greece; ppapaporf@for.auth.gr (P.K.P.); esarroy@gmail.com (E.S.); leleav_av@yahoo.gr (E.A.)
* Correspondence: eabraham@for.auth.gr; Tel.: +30-2310-992301

**Abstract:** *Sideritis scardica (S. scardica)* is an endemic medicinal species of the Central Balkan Peninsula. The aerial parts are traditionally used in folk medicine and, therefore, have been collected extensively from natural habitats. Overexploitation in combination with climate change has resulted in severely fragmented populations. In this context, the purpose of this study was to access the abundance and phenotypic diversity of *S. scardica* populations in relation to plant community structure and environmental and anthropogenic factors in six mountainous areas of Northern Greece. For this reason, the floristic composition and diversity was determined by accessing the number of plant species, number of individuals per plant species, and plant cover in each study area. In addition, the soil properties of the studied areas were determined and the phenotypic diversity of *S. scardica* populations was accessed through the imaging of leaf and inflorescence main characteristics. As a result, 141 plant species were identified in all studied areas, while the floristic composition clearly distinguished the North-Central from the North-Eastern studied areas. *S. scardica* was the predominant species in the habitats where the presence of forbs was favored, while a high presence of graminoid and shrub species in the study areas depressed its presence. A high coefficient of variations was recorded among the six populations, varying from 12.2%–29.2% and 13.3%–43.1% for inflorescence and leaf traits.

**Keywords:** Greek mountain tea; medicinal species; endemic populations; floristic diversity; morphological variation; plant community structure; grazing

---

## 1. Introduction

*Sideritis scardica (S. scardica)* is an endemic species of the Central Balkan Peninsula and its distribution is restricted to Bulgaria, Greece, North Macedonia, Albania, Serbia, and Turkey [1,2]. In Greece, the species is distributed mostly to the North and North-Eastern parts of the country. The plants of *S. scardica* are collected from natural habitats, as they are traditionally used in medicine (for herbal preparations, infusions, decoctions, etc.) for treating gastrointestinal disorders, sore throats, asthma, bronchitis, the common cold, etc. The reported bioactivities of hydro-alcoholic and other extracts—i.e., anti-inflammatory, neuroprotective, analgesic, antioxidant, antimicrobial, as well as its gastrointestinal protection properties—are attributed to a variety of secondary metabolites of *S. scardica* belonging to terpenes, di- and triterpenes, phenylpropanoids, and polymethoxylated flavones [3–8]. Due to the late reports on such bioactivities, *S. scardica* aroused considerable interest as a medicinal plant and its natural populations have been over-exploited over the past decade. This, in combination with

environmental (climate change) and management factors (intensity of grazing), resulted in severely fragmented populations, which characterizes the species as near-threatened (International Union for Conservation of Nature (IUCN) red list) and, therefore, it is under special regime for conservation and use [9].

The abundance of a specific species in a habitat is mainly regulated by environmental factors and by interactions with other species—i.e., the composition and structure of the plant community. Environmental parameters, such as the topography, climate, soil, and management (i.e., grazing, collections, etc.) [10] of the habitats are the key factors that shape the structure of plant communities, which, in turn, could be a determining factor for the growth of *S. scardica* populations [11,12]. Generally, *S. scardica* occurs naturally in open, dry, stony places of subalpine and alpine vegetation belts. Additionally, environmental factors could affect the dispersal of a species. Particularly, an investigation on the main reproductive biology parameters of *S. scardica* showed that seed viability and germination is highly dependent on the peculiarities of the climatic conditions during the seed maturity at harvest point [13].

Phenotypic diversity in plants is required for populations to evolve in response to environmental changes, and its maintenance is crucial for long-term species survival [14]. Therefore, knowledge of the phenotypic variation of an endangered species under different environments is the prerequisite for understanding its genetic variation pattern, fitness, and evolutionary capacity to adapt to environmental changes, and it is crucial for their in situ conservation and management. It has been documented that one of the predicted consequences of global climate change is the movement of plant species to higher latitudes from the climate they were so far adapted [15]. During this migration process towards higher altitudes, a drastic decrease—or even extinction—of the plant species could be a consequence. As a result of this migration, adaptive—but also random evolutionary processes—and plastic responses to growth conditions seem to be crucial for the survival of plants while shaping the phenotypic variability of species such as *S. scardica*, especially in the alpine landscapes. Plasticity is considered to be a dominant source of phenotypic variations in plants, as it affects the natural selection and, consequently, the patterns of diversification among populations and, ultimately, species [16]. Such diversification arose from changes in the individual's behavior at a morphological and/or physiological level under different environmental conditions, while such changes may be crucial to survival in heterogeneous and variable conditions [17,18]. Thus, as a result, plasticity promotes evolutionary diversification if the produced phenotypes provide adaptive diversity that, under selection, becomes evolutionarily fixed, which is supported also from a genetic point of view [16,19]. Information on the diversity of *S. scardica* populations is still unexplored. The only survey performed on nine *S. scardica* populations from Northern Macedonia and Greece demonstrated relatively low intra-population diversity and significant inter-population differentiation, with gene diversity higher for the South-Eastern populations, as recorded through amplified fragment length polymorphism (AFLP) marker analysis [20]. In addition, a positive significant correlation between genetic and geographical distances was reported, which revealed a pattern of isolation by distance, and further confirmed limited gene flow between analyzed populations, which facilitates the establishment of local adaptations.

As reported by Yordanova and Apostolova [21], anthropogenic activities—i.e., collections and trampling—have a significant impact on *S. scardica* abundance. According to the higher density of *S. scardica* plants in habitats where *Juniperus communis* was present, it was hypothesized that *J. communis* have potentially offered protection. For instance, on the Greek mountain Olympus, which is usually highly accessible to tourists, the *S. scardica* plants are more susceptible to risk of collection, as they can be easily observed [22]. Furthermore, an additional pressure factor is the habitat loss from humans through land use changes, ploughing, and the creation of forest trails, as well as grazing pressure, which could affect the species' abundance and diversity [21].

Despite the high importance as medicinal plant species, there is restricted information about the *S. scardica* habitat characteristics. In this regard, the management and conservation of vulnerable plant species, such as *S. scardica*, requires knowledge about how their populations are affected by

abiotic (environmental factors) and biotic (plant community, herbivores, etc.) factors, as well as human activities.

The aim of the present study was to access the abundance and the phenotypic diversity of *S. scardica* populations in relation to floristic composition, environmental factors (slope, exposure, and soil), and anthropogenic management factors (overharvesting and grazing) in six mountainous sites of Northern Greece. A thorough understanding of its habitat will highlight the present and future threats and will guide its protection through sustainable management. This, in combination with the assessment of the phenotypic diversity of the species, could potentially substitute the basis for plant selection in terms of breeding purposes, and also sustainable cultivation as an alternative of ex situ preservation.

## 2. Materials and Methods

### 2.1. The Studied Species

*Sideritis scardica* Griseb. (synonyms: *Sideritis florida* (Boiss. & Heldr.), *Sideritis raeseri* subsp. *florida* (Boiss. & Heldr.; Papan. & Kokkini), and *Sideritis scardica* subsp. *longibracteata* (Papan. & Kokkini) is a perennial herbaceous plant that belongs to the Lamiaceae family [23–25]. *S. scardica* is a cross-pollinated species with stems up to 40 cm high, branched and/or unbranched, opposite leaves covered by gray hairs, inflorescences with dense spikes, middle bracts 12–20 mm long, lemon yellow glandular corolla, and calyx tubular–campanulate [26]. The species is reported as diploid, with 2n = 32 [27], although in some materials, chromosomal complements of 2n = 32 + 0 2B and 2n = 30 have also been discovered [28]. In natural habitats, *S. scardica* is mostly reproduced through seeds, while in ideal environmental conditions (i.e., temperature, humidity, soil texture, etc.), they are also reproduced through offshoot clonal propagation.

### 2.2. Study Area

The study was conducted in the mountainous areas of Northern Greece (prefecture of North-Central and North-Eastern Macedonia, Greece) (Figure 1), in which the study sites were identified in collaboration with local people and local forest services. Permission was asked from the Ministry of Environment and Energy in Greece so as to collect the samples legally for research activities. Six case study areas were selected, in which *Sideritis scardica* Griseb. (known as Greek mountain tea, *Lamiaceae*) is regarded as endemic; these areas were: Komnina (VerKom), Tetralofos (VerTet), Olympos (Oly), Falakro (Fal), Paggaio (Pag), and Menoikio (Men). The first two aforementioned areas are situated in Mountain Vermio, both of which belong to North-Central Greece in terms of the vegetation zone or phytosociological aspect, as well as the Olympos region [28,29]. The three final aforementioned areas belong to North-Eastern Greece.

All study areas, except for Komnina and Tetralofos, belong to Special Areas of Conservation (SAC) of the Natura 2000 network, or even to Special Protection Areas (SPA), in Northern Greece (Table 1) [30]. Moreover, all study areas are regarded as pseudo-alpine grasslands, apart from Komnina, which is regarded as a shrubland area. The habitat type 6170–"alpine and subalpine calcareous grasslands" [31] is the most dominant habitat type for Olympos, Paggaio, and Menoikio. The habitat type 62A0–"Eastern sub-Mediterranean dry grasslands (Scorzoneratalia villosae)" appeared in Falakro. *S. scardica* is a typical species of both habitat types.

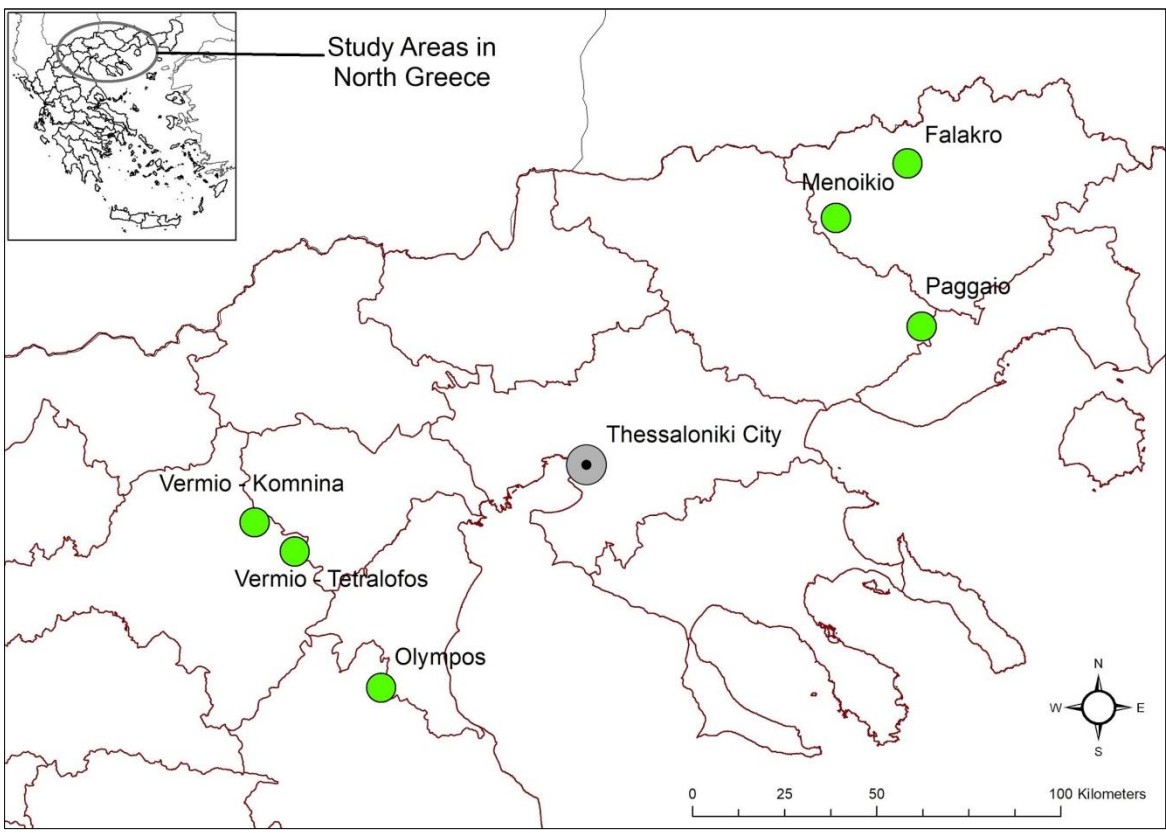

**Figure 1.** The green markers show the location of six case study areas in Northern Greece.

**Table 1.** Habitat types, Natura 2000 area categories, and dominant species for each study area.

| Study Areas | Habitats | Habitat Types | Code | Natura 2000-Area Category | Dominant Species |
|---|---|---|---|---|---|
| Komnina (VerKom) | Shrubland | - | GR1210001 | SAC [1] | *Helianthemum canum* ssp. *canum, Fumana procumbens, Sesleria caerulea* |
| Tetralofos (VerTet) | Pseudo-alpine grassland | - | GR1210001 | SAC | *Pilosella hoppeana* ssp. *testimonialis, Thymus longicaulis* ssp. *longicaulis, Anthyllis vulneraria* ssp. *rubriflora* |
| Olympos (Oly) | Pseudo-alpine grassland | 6170 | GR1250001 | SAC and SPA [2] | *Sesleria* sp., *Festuca ovina, Lotus corniculatus* |
| Falakro (Fal) | Pseudo-alpine grassland | 62A0 | GR1140004 | SAC | *Festuca ovina, Trifolium heldreichianum, Helianthemum canum* ssp. *canum* |
| Paggaio (Pag) | Pseudo-alpine grassland | 6170 | GR1150005 and GR1150011 | SAC and SPA | *Galium mollugo* aggr., *Festuca ovina, Trifolium heldreichianum* |
| Menoikio (Men) | Pseudo-alpine grassland | 6170 | GR1260004 and GR1260009 | SAC and SPA | *Festuca* sp. 2, *Fragaria vesca, Potentilla cinerea* |

[1] SAC: Special Areas for Conservation, [2] SPA: Special Protection Areas.

The case study areas lie at a range between 1047 m above sea level in Komnina to 2042 m in Olympos (Table 2). Initially, each area was visited, and continuous populations of the species were identified in collaboration with the local forestry services. The size of the area where these populations were identified ranged from 5 to 6 ha. Environmental factors, such as aspect and slope, were recorded, as well as plant cover, grazing intensity, and illegal collection of *S. scardica*. The forage utilization percentage (FUP) was estimated by the Ocular estimate-by-plot method [32] as an indicator of grazing intensity. Generally, the FUP was low to moderate, with less than 50% in all cases. There was no grazing in the areas of Komnina or Tetralofos (Table 2). The forest service issues annual regulations for the collection of *S. scardica*. However, people collect it illegally for personal use and/or trade. The illegal collection of *S. scardica* was recorded by interviewing local people and forest service staff. Komnina and Olympos were the areas with the highest collection pressure, while there was no collection pressure in Falakro or Menoikio, which were relatively inaccessible sites and far from inhabited areas (Table 2). As for mean annual temperature and mean annual precipitation, the data were downloaded by using the peer reviewed article of Karger et al. (2017) [33].

**Table 2.** General location, environmental variables, management, and soil features recorded in each study area and used for the multivariate analyses.

| Study Areas | VerKom | VerTet | Oly | Fal | Pag | Men |
|---|---|---|---|---|---|---|
| Altitude (m asl) | 1047 | 1703 | 2042 | 1958 | 1381 | 1696 |
| Latitude | 40°35′ 51.33″ | 40°27′ 57.55″ | 40°2′ 54.36″ | 41°18′ 3.42″ | 40°56′ 13.68″ | 41°11′ 33.11″ |
| Longitude | 21°48′ 58.04″ | 21°59′ 51.07″ | 22°20′ 57.13″ | 24°4′ 45.82″ | 24°4′ 49.72″ | 23°45′ 46.28″ |
| Explanatory Variables | | | | | | |
| Mean annual precipitation (mm) | 662 | 720 | 532 | 822 | 816 | 769 |
| Mean annual temperature (°C) | 10 | 6.3 | 4 | 3.7 | 7.4 | 5.7 |
| Management | | | | | | |
| Plant Cover % | 45.5 | 65.0 | 45.3 | 48.3 | 63.0 | 72.3 |
| Grazing [1] | 0 | 0 | 1 | 1 | 1 | 1 |
| Collection [2] | 1 | 1 | 1 | 0 | 1 | 0 |
| Soil features | | | | | | |
| MAP (mm) [3] | 662 | 720 | 532 | 822 | 816 | 769 |
| MAT (°C) [4] | 10 | 6.3 | 4 | 3.7 | 7.4 | 5.7 |
| Pl C (%) [5] | 45.5 | 65 | 45.33 | 48.33 | 63 | 72.33 |
| pH | 8.1 | 7.9 | 7.7 | 7.15 | 6.8 | 7.9 |
| OM (%) [6] | 7 | 7 | 8 | 7 | 7.7 | 4.8 |
| CaCO$_3$ (%) | 4 | 4.4 | 25.5 | 9.8 | 4 | 6.3 |
| N | 43.17 | 40 | 199 | 16 | 73.13 | 48.6 |
| P | 33.73 | 30 | 10.6 | 22 | 38.59 | 4.16 |
| *S. scardica* (individuals/m$^2$) | 0.83 | 1 | 0.83 | 3.67 | 1.17 | 4.33 |

[1] Grazing: Grazing status according to direct observations in the field and forage utilization use (FUP): presence: 1 (FUP > 20%) or absence: 0 (FUP < 20%), [2] Collection: illegal collection by people for personal use or illegal trade: presence: 1 or absence: 0, [3] MAP: monthly average precipitation, [4] MAT: monthly average temperature, [5] Pl C: plant cover, [6] OM: organic matter.

## 2.3. Analysis of Soil Characteristics

Six soil samples (from depth < 30 cm according to the root system of *S. scardica*) were taken from each study site (close to the quadrats for vegetation composition) and transferred in the laboratory. The samples were placed in trays, allowed to dry at an ambient temperature (32–35 °C), grinded in mortar, and sieved in order to obtain particles with the diameter of ≤ 2 mm. The soil pH was measured in a water-saturated soil sample (the ratio of distilled water:soil was 1:1). The soil organic matter content was recorded spectrophotometrically at 600 nm by the use of $K_2Cr_2O_7$ and $H_2SO_4$. $KHC_8H_4O_4$

was used as standard in order to generate a calibration curve. The content of boron (B) in the soil samples was determined by the azomethine–H method [34], using the double beam spectrophotometer HITACHI U-2900 IIO UV (Hitachi, Ltd., Tokyo, Japan). Total nitrogen (N) was determined through the Kjeldahl method with the Vapodest 50 system (Gerhardt, Königswinter, Germany). The quantitative determination of soil macronutrients was performed by an inductively coupled plasma optical emission spectrometry (ICP-OES) system (Perkin Elmer Optima 2100DV). The $CaCO_3$ soil was determined by the use of Bernards' apparatus after the collection and measurement of the released $CO_2$, after the addition of hydrochloric acid on soil carbonates.

## 2.4. Diversity Data

Plant cover, the number of plant species, and the number of individuals per plant species, were recorded in order to estimate vegetation composition and floristic diversity [35,36] in each region during the flowering period in July, 2018. Six quadrats (1mX1m) were randomly placed in each of the studied regions. The specimens were collected in order to be identified in the laboratory. Species identification was performed according to Tutin et al. (1968–80; 1993) [37,38], Strid and Tan (1991) [28], Strid (1986) [29], and Pignatti (1997) [39]. Nomenclature followed Euro+Med (2006) [40]. A floristic bibliography referring to these regions is included in: "Mountain Flora of Greece" [21,22] and "Flora Europaea" [37,38]. The recorded plant species were grouped into graminoids (G), legumes (L), forbs (F), and woody (W) species.

Moreover, the species number, the diversity indices of Shannon–Wiener, Simpson, Margalef, equitability, evenness, Berger–Parker, and McIntosh [41,42], as well as a Jaccard index, were estimated for each population. The Jaccard index was estimated by the following formula:

$$Cj = j/(a + b - j) \tag{1}$$

where: j = the number of species common to both sites, a = the number of species in site A, and b = the number of species in site B.

## 2.5. Evaluation of Phenotypic Characteristics

A phenotypic evaluation was conducted in twenty-four *S. scardica* individuals per population (for OLY, only 16 were found) at the stage of full blooming. The maximum stem length and the number of stems per plant were recorded on an individual basis in situ for the six study sites. All stems were collected separately for each individual, transferred in plastic containers to the laboratory, and hydrated for 24 h in water. The evaluation of the basal leaf and inflorescence morphological characteristics referring to area, perimeter, length, and width was conducted through Image Pro Plus Software (Media Cybernetics).

## 2.6. Statistical Analysis

All data concerning plant cover, species composition, floristic diversity, and phenotypic evaluation were analyzed statistically by using a one-way ANOVA in SPSS ver. 25 for Windows [43]. For the means comparisons, a Tukey's HSD (honestly significant difference) test and standard error (SE) were used at $P \leq 0.05$ to establish significant differences. In cases where the number of replicates was not equal, a Welch test was also applied in order to evaluate the significance of the ANOVA analysis.

A detrended correspondence analysis (DCA) was used for the selection of the appropriate canonical ordination. As the length of the first DCA axis—which was indicative of the beta-diversity—was higher than 3 and lower than 4 SD (3.5149), the samples seemed to be rather heterogeneous, for which simple linear methods were not suitable. For that reason, a transformation-based principal component analysis (tb-PCA), and a transformation-based canonical redundancy analysis (tb-RDA) were performed using the Hellinger transformation. In both analyses, only the taxa that were presented to all plots in each site, and they had a percentage of more than 5% in species composition were included. Prior to the

analyses, all the data were logarithmically transformed. The tb-RDA was carried out for the ordination of floristic composition constrained by the explanatory variables presented in Table 2. An automatic stepwise forward model building method based on the adjusted R2 and P-value was used for the identification of significant constrains (function ordi2step). While only the significant constrains were used during the calculation of the constrained tb-RDA, the rest of the constrains were fitted on the graph in a second step (function envfit). All the analyses was carried out by using the package Vegan (v2.5-6) of R.

An additional PCA analysis was conducted in order to study the patterns of variation in the data sets of phenotypic evaluation. The PCA output was used to construct biplots to visualize the distribution (and clustering) of *S. scardica* populations in relation to the morphometric traits. The biplots helped in the identification of the clusters of morphometric traits that could be related mostly to each population discrimination, thus, causing its clustering.

## 3. Results

### 3.1. Floristic Composition and Diversity of the Studied Areas

The floristic composition significantly differed in the studied areas. The areas of Olympos and Falakro had a significantly higher percentage of graminoids and legumes in plant cover, but a lower percentage of forbs (Table 3) compared to the other areas. On the other hand, the areas of Tetralofos, Paggaio, and Menoikio had a lower percentage of graminoids, but higher of forbs. Finally, the lower percentage of legumes were recorded in the areas of Komnina and Menoikio, while the higher were in woody species and the *S. scardica* were in Olympos and Menoikio, respectively (Table 3). Regarding the number of species per functional group, only the number of forbs and woody species significantly differed among the studied areas. In particular, the highest number of forbs and woody species were recorded in Tetralofos and Olympos, respectively (Table 3).

**Table 3.** Percentage (%) of different functional groups, number of different plant species per functional group, and number of individual plants of *Sideritis scardica* per m$^2$ in each study area.

| Study Areas | G [1] % | G [1] No [6] | L [2] % | L [2] No [6] | F [3] % | F [3] No [6] | W [4] % | W [4] No [6] | S [5] % | S [5] No [7] |
|---|---|---|---|---|---|---|---|---|---|---|
| VerKom | 29.2ab | 2.3a | 5.7b | 1.2a | 58.2a | 7.5bc | 5.8ab | 1.2ab | 1.1b | 0.8b |
| VerTet | 13.2c | 3.3a | 20.1ab | 3.2a | 52.1ab | 11.8a | 13.8ab | 1.8ab | 0.8b | 1.0b |
| Oly | 40.5a | 2.7a | 12.3ab | 1.2a | 29.3b | 5.8c | 16.4a | 2.5a | 1.4b | 0.8b |
| Fal | 30.9ab | 1.7a | 27.3a | 2.8a | 31.9b | 6.8bc | 6.3ab | 0.8b | 3.7ab | 3.7ab |
| Pag | 23.4bc | 2.5a | 15.3ab | 2.0a | 56.3a | 9.0abc | 3.6ab | 1.7ab | 1.4b | 1.2b |
| Men | 23.2bc | 1.7a | 6.1b | 1.3a | 61.8a | 10.2ab | 3.2b | 1.2ab | 5.8a | 4.3a |

[1] G: graminoids, [2] L: legumes, [3] F: forbs, [4] W: woody, [5] S: *Sideritis scardica*, [6] No: number of different plant species per functional group in each study area, [7] No: number of individual plants of *Sideritis scardica* per m$^2$, which were tested by using the Tukey's HSD (honestly significant difference) test. Different letters in the same column indicate significant differences among the six study areas ($p \leq 0.05$).

In detail, 141 species in total were recorded in the studied areas (supplementary Table S1). According to the PCA, the floristic composition clearly distinguished the studied areas, with the North-Central areas (Komnina, Tetralofos, and Olympos) on the left side of the PC1 axis and the North-Eastern areas (Falakro, Paggaio, and Menoikio) on the right (Figure 2). PC1 and PC2 accounted for 58.5% and 41.5%, respectively, of the total variation. The plant species formed four distinct clusters (Figure 2). The species in the green cluster characterized the vegetation in Komnina, while the species in the blue cluster characterized the North-Eastern areas and mainly the area of Menoikio. The species in the purple cluster mainly characterized the vegetation of Olympos, but they were also broadly distributed in both North-Central and Eastern areas. Finally, the species in the pink cluster, which included *S. scardica*, mainly characterized the species that were predominant in both the North-Central

and North-Eastern areas. In accordance with the PCA, the highest Jaccard similarity index was among the North-Central areas and the Eastern areas (Table 4). Furthermore, the lowest Jaccard similarity index (Table 4) indicated that low similarity (0.04) was between Komnina and Paggaio.

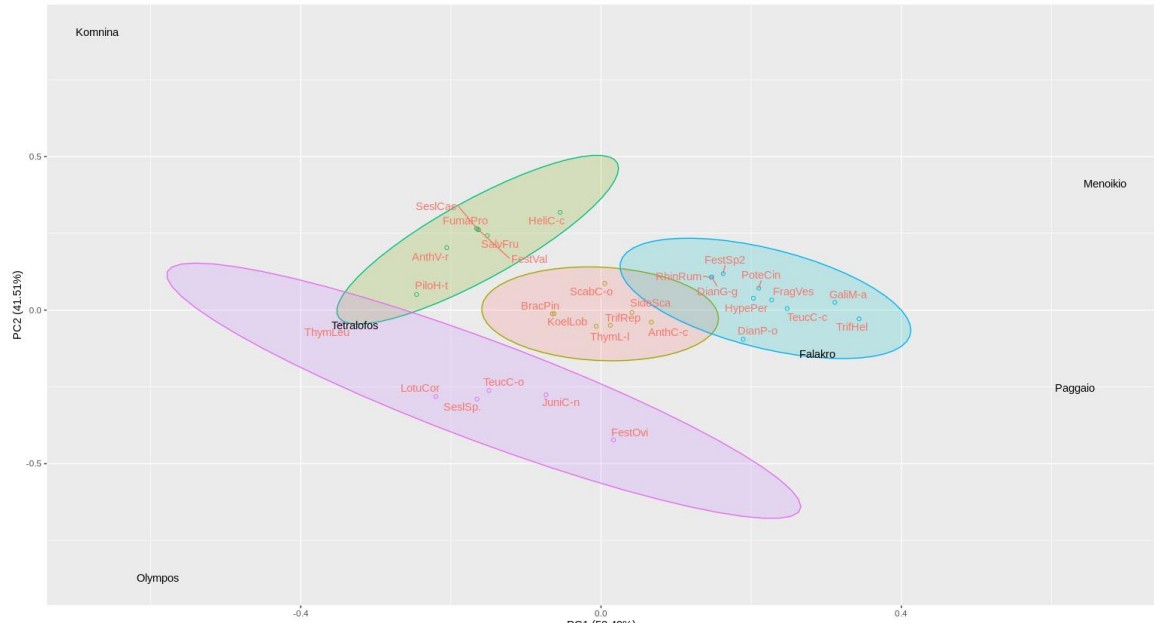

**Figure 2.** Principal component analysis (PCA) of the studied areas based on floristic composition. SeslCae: *Sesleria caerulea*, FumaPro: *Fumana procumbens*, HeliC-c: *Helianthemum canum* ssp. *canum*, AnthV-r: *Anthyllis vulneraria* ssp. *rubriflora*, PiloH-t: *Pilosella hoppeana* ssp. *testimonialis*, FestVal: *Festuca valesiaca*, SalvFru: *Salvia fruticosa*, ScabC-o: *Scabiosa columbaria* ssp. *ochroleuca*, BracPin: *Brachypodium pinnatum*, KoelLob: *Koeleria lobata*, ThymL-l: *Thymus longicaulis* ssp. *longicaulis*, TrifRep: *Trifolium repens*, SideSca: *Sideritis scardica*, AnthC-c: *Anthemis cretica* ssp. *columnae*, FestSp2: *Festuca* sp. 2, RhinRum: *Rhinanthus rumelicus*, DianG-g: *Dianthus gracilis* ssp. *gracilis*, PoteCin: *Potentilla cinerea*, HypePer: *Hypericum perforatum*, DianP-o: *Dianthus petraeus* ssp. *orbelicus*, FragVes: *Fragaria vesca*, TeucC-c: *Teucrium chamaedrys* ssp. *chamaedrys*, GaliM-a: *Galium mollugo* aggr., TrifHel: *Trifolium heldreichianum*, ThymLeu: *Thymus leucotrichus*, LotuCor: *Lotus corniculatus*, SeslSp.: *Sesleria* sp., TeucC-o: *Teucrium chamaedrys* ssp. *olympicum*, JuniC-n: *Juniperus communis* ssp. *nana*, and FestOvi: *Festuca ovina*.

**Table 4.** Values of Jaccard similarity index between the study areas.

| Study Areas | VerKom | VerTet | Oly | Fal | Pag | Men |
|---|---|---|---|---|---|---|
| VerKom | 1 | | | | | |
| VerTet | 0.115385 | 1 | | | | |
| Oly | 0.061225 | 0.140845 | 1 | | | |
| Fal | 0.102041 | 0.136986 | 0.066667 | 1 | | |
| Pag | 0.044776 | 0.125 | 0.066667 | 0.1 | 1 | |
| Men | 0.096154 | 0.075 | 0.0625 | 0.152174 | 0.189655 | 1 |

The tb-RDA analysis revealed that the floristic composition of the studied habitats was affected significantly by the altitude and the levels of precipitation. For instance, the presence of forbs was favored in habitats characterized from high precipitation levels in comparison with graminoids and woody species. Likewise, in higher altitudes, the presence of graminoids and shrubs was favored (Figure 3). Grazing did not seem to be a statistically significant factor; however, in the studied areas that were grazed, it was observed that the forbs were mostly developed, in contrast with the un-grazed, where the presence of woody was favored (Figure 3). The presence of *S. scardica* was not a significant variable for the floristic composition. Nevertheless, a high presence of grasses, such as *Festuca ovina* and

*Sesleria* sp., and shrubs in the study areas where *S. scardica* did not exist were observed, while, in contrast, the *S. scardica* was dominant in the habitats where the presence of forbs was favored. Eventually, in the study sites where *S. scardica* was present, no severe collection phenomena were recorded.

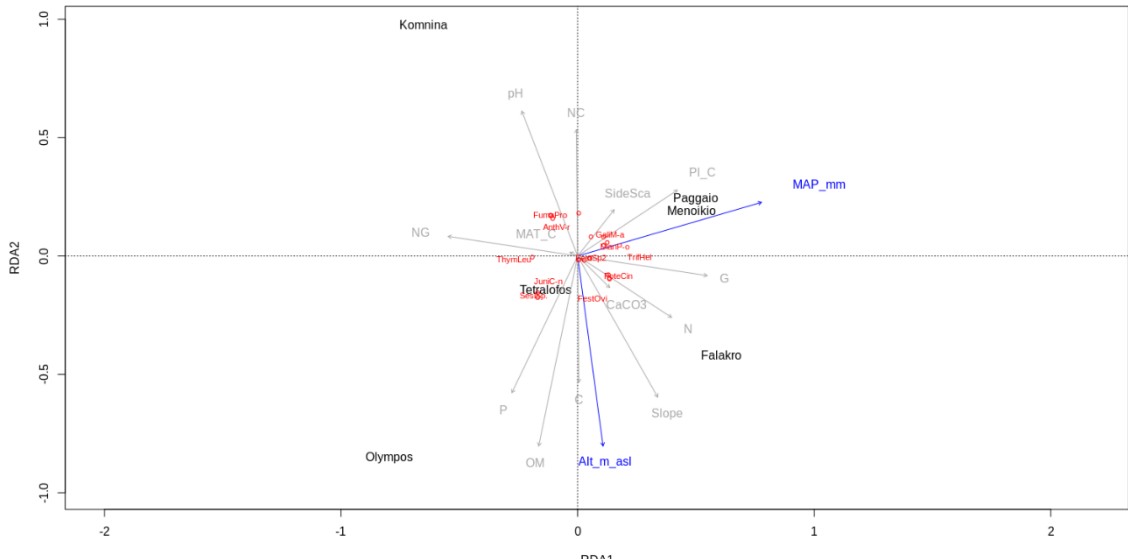

**Figure 3.** Transformation-based canonical redundancy analysis (tb-RDA) diagrams of floristic composition in the studied sites. Axis 1 and 2 explain 54% and 41% of the floristic composition, respectively. Explanatory variables are represented by arrows, and the significant ones ($p \leq 0.05$) are blue. Abbreviations of explanatory variables and plant species: SideSca: *Sideritis scardica*, FumaPro: *Fumana procumbens*, AnthV-r: *Anthyllis vulneraria* ssp. *rubriflora*, GaliM-a: *Galium mollugo* aggr., DianP-o: *Dianthus petraeus* ssp. *orbelicus*, FestSp2: *Festuca* sp. 2, TrifHel: *Trifolium heldreichianum*, PoteCin: *Potentilla cinerea*, FestOvi: *Festuca ovina*, SeslSp.: *Sesleria* sp., JuniC-n: *Juniperus communis* ssp. *nana*, and ThymLeu: *Thymus leucotrichus*.

The higher floristic diversity in terms of Shannon–Wiener, Species Number, and Margalef indices was recorded in the Tetralofos area (Table 5) in comparison with the other studied areas. Nevertheless, the floristic diversity in terms of Simpson, equitability, Berger–Parker, and the McIntosh index (supplementary Table S2) presented no significant differences among the study areas.

**Table 5.** Mean values of species number and the diversity indices of Shannon–Wiener, Simpson, Margalef, equitability, Berger–Parker, and McIntosh, which were tested by using the Tukey's HSD test.

| Study Areas | Shannon-Wiener | Simpson | Species Number | Margalef | Equitability | Berger-Parker | McIntosh |
|---|---|---|---|---|---|---|---|
| VerKom | 2.06b | 6.87a | 12.17b | 2.60b | 0.61a | 0.28a | 0.67a |
| VerTet | 2.55a | 10.73a | 20.17a | 3.89a | 0.63a | 0.22a | 0.73a |
| Oly | 2.16ab | 8.00a | 12.17b | 2.74b | 0.69a | 0.24a | 0.71a |
| Fal | 2.04b | 6.65a | 12.17b | 2.46b | 0.63a | 0.32a | 0.64a |
| Pag | 2.40ab | 10.70a | 15.17b | 3.25ab | 0.65a | 0.20a | 0.75a |
| Men | 2.29ab | 8.62a | 14.33b | 2.91b | 0.69a | 0.22a | 0.72a |

Different letters in the same column indicate significant differences among the six areas ($p \leq 0.05$).

### 3.2. Evaluation of the Phenotypic Diversity among the Populations in the Studied Areas

In the present study, an attempt was made to evaluate the phenotypic diversity concerning the morphometric traits of basal leaves and inflorescences of *S. scardica* individuals among the six study areas, and our findings are presented in Table 6. The maximum stem length of the plants varied between 11.4 cm and 38.2 cm among the six mountainous areas. The populations from Olympos (Oly) and Falakro (Fal) represented the shorter stem length compared to the rest of the populations, where a

2 to 3 times higher stem length was recorded. The statistical analysis revealed significant differences on the maximum stem length of the plants, with the population from Komnina (VerKom) exhibiting a significantly greater maximum stem length. In contrast with the stem length, the population from Falakro (Fal) expressed a greater number of stems per individual, differing significantly from all the other investigated populations (VerTet, Oly, Pag, and Men), except for the one from Komnina (VerKom).

In regard to the morphometric traits of inflorescence, significant variations were observed among the six populations, with the area varying between 6.1–16.2 cm$^2$, the perimeter from 13.4–28.0 cm, the length from 3.5–8.7 cm, and the width from 2.8–3.0 cm. According to the statistical evaluation of the morphometric traits, the population VerKom represented superior characteristics compared to the other populations, while these were considered almost 2-fold greater in area, perimeter, and length of inflorescence from the population Oly and Fal. Although the population of Fal (together with Oly) exhibited smaller inflorescence characteristics, they represented a significantly greater leaf area (3.6 cm$^2$), perimeter (13.1 cm), and length (6.3 cm) compared to the populations VerKom, VerTet, and Oly (except the leaf perimeter). A relatively high coefficient of variations was recorded among the six populations, varying from 12.2%–29.2% and 13.3%–43.1% for inflorescence and leaf traits, respectively.

The biplot of the PCA analysis, based on the leaf and inflorescence morphometric characteristics, explained 93.45% of the total variation, calculated from Jaccard's coefficient (Figure 4). According to the biplot of PCA, the six *S. scardica* populations represented a clear clustering in correlation to the phenotypic traits. Specifically, the populations Fal, Pag, and Men were grouped in the same cluster, and were highly positively correlated with leaf perimeter and length. The populations Oly, VerTet, and VerKom were classified on their own in different clusters, with the latter highly correlated with the perimeter and length of inflorescence. Nevertheless, the distribution of the individuals per population, according to the PCA analysis of the phenotypic traits (supplementary Figure S1), revealed highly variable populations, such as VerKom, Oly, Pag, and Fal. In contrast, the populations of VerTet and Men seemed to be more compact as their individuals clustered together closely.

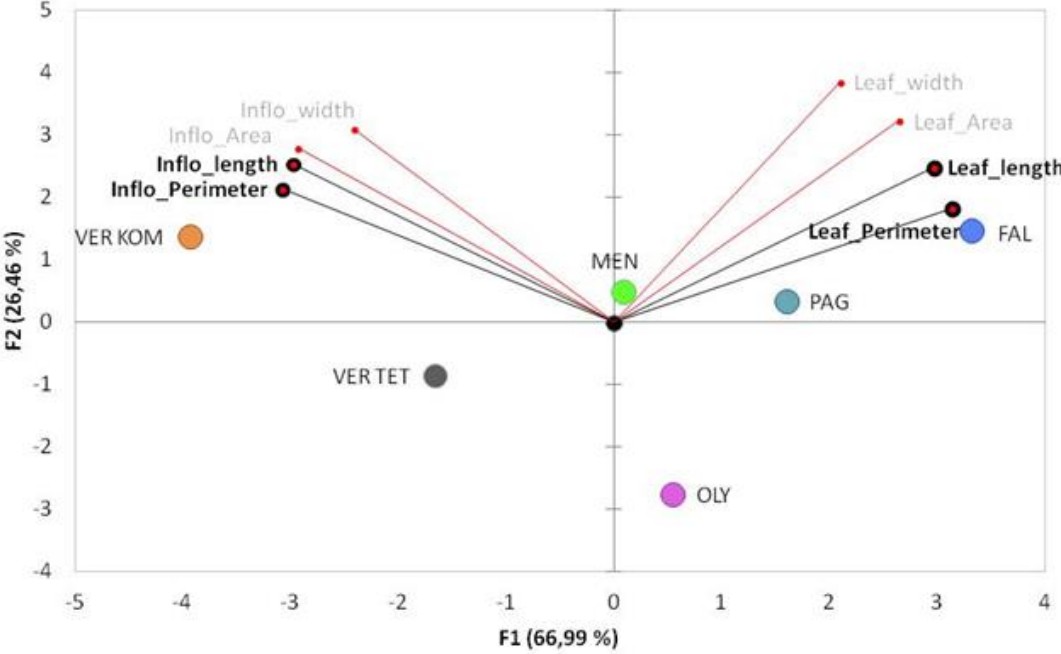

**Figure 4.** Biplot of principal component analysis of six *Sideritis scardica* populations based on leaves and inflorescence morphometric traits. Inflo_Perimeter: inflorescence perimeter; Inflo_length: inflorescence length; Inflo_Area: inflorescence area; Inflo_width: inflorescence width; VERKOM: Komnina; VERTET: Tetralofos: OLY: Olympus; PAG: Paggaio; MEN: Menoikio; FAL: Falakro.

**Table 6.** In situ phenotypic evaluation of wild plants (in terms of inflorescence and leaves) from six native *Sideritis scardica* populations of Northern Greece.

| | | | | | | | | | | |
|---|---|---|---|---|---|---|---|---|---|---|
| | | | | | *Inflorescence* | | | | | |
| **Population** | **Individuals Number** | **Max. Stem length** | **Area (cm$^2$)** | **CV%** | **Perimeter (cm)** | **CV%** | **Length (cm)** | **CV%** | **Width (cm)** | **CV%** |
| VerKom | 24 | 38.2 ± 1.7 a | 16.2 ± 0.7 a | 20.2 | 28.0 ± 1.3 a | 22.2 | 8.7 ± 0.4 a | 19.5 | 3 ± 0.1 | 13.2 |
| VerTet | 24 | 27.3 ± 1.7 b | 10.7 ± 0.6 b | 28.8 | 21.2 ± 0.7 b | 15.3 | 6.2 ± 0.2 b | 18.3 | 3 ± 0.1 | 19.1 |
| Oly | 16 | 11.4 ± 0.7 c | 6.1 ± 0.4 d | 29.2 | 13.4 ± 0.7 c | 20.8 | 3.5 ± 0.2 d | 20.5 | 2.8 ± 0.1 | 12.7 |
| Pag | 24 | 26.8 ± 1.3 b | 9.0 ± 0.5 bc | 27.9 | 15.7 ± 0.8 c | 25.2 | 5.0± 0.3 bc | 27.8 | 2.8 ± 0.1 | 13.5 |
| Men | 24 | 27.9 ± 0.9 b | 10.3 ± 0.6 b | 27.6 | 16.5 ± 0.6 c | 18.3 | 5.2 ± 0.2 bc | 20.8 | 3 ± 0.1 | 12.2 |
| Fal | 24 | 14.4 ± 0.7 c | 7.2 ± 0.2 cd | 16.9 | 14.5 ± 0.4 c | 12.4 | 4.1 ± 0.1 cd | 12.3 | 2.9 ± 0.1 | 13.5 |
| | | | | | *Leaves* | | | | | |
| **Population** | **Individuals number** | **No. of stems/plant** | **Area (cm$^2$)** | **CV%** | **Perimeter (cm)** | **CV%** | **Length (cm)** | **CV%** | **Width (cm)** | **CV%** |
| VerKom | 24 | 4.2 ± 0.7 ab | 2.6 ± 0.2 b | 43.1 | 10.3 ± 0.5 b | 24.5 | 4.7 ± 0.3 b | 29.6 | 0.9 ± 0 | 25.8 |
| VerTet | 24 | 3.6 ± 0.6 b | 2.7 ± 0.2 b | 42.2 | 10.2 ± 0.3 b | 16.3 | 4.8 ± 0.2 b | 16.6 | 0.9 ± 0 | 18.1 |
| Oly | 16 | 2.9 ± 0.7 b | 2.5 ± 0.2 b | 36.1 | 11.2 ± 0.7 ab | 24.6 | 4.9 ± 0.3 b | 23.7 | 0.9 ± 0.1 | 22.7 |
| Pag | 24 | 3.6 ± 0.7 b | 3.1 ± 0.2 ab | 35.5 | 11.8 ± 0.5 ab | 18.5 | 5.6 ± 0.2 ab | 19.4 | 1 ± 0 | 19.2 |
| Men | 24 | 3.7 ±0.4 b | 2.9 ± 0.3 ab | 43.2 | 11.7 ± 0.6 ab | 23.2 | 5.1 ± 0.3 b | 24.2 | 1 ± 0 | 20.3 |
| Fal | 24 | 6.5 ± 0.6 a | 3.6 ± 0.2 a | 25.4 | 13.1 ± 0.4 a | 13.2 | 6.3 ± 0.2 a | 13.3 | 1 ± 0 | 16.9 |

VerKom: Komnina; VerTet: Tetralofos: Oly: Olympus; Pag: Paggaio; Men: Menoikio; Fal: Falakro. Values within the columns correspond to means ± standard error. The replicates per population were n = 24 individual plants per population (except Oly, where n = 16). CV% refers to coefficient of variation for each variable evaluated. Different letters within the columns indicate statistically significant differences according to Tukey's HSD (honestly significant difference) test for $p \leq 0.05$.

## 4. Discussion

In the present study, the floristic composition was analyzed in six mountainous areas that were habitats of *S. scardica,* a medicinal plant whose natural populations are endangered by over-exploitation. The floristic composition clearly distinguished the North-Central areas (Komnina, Tetralofos, and Olympos) and North-Eastern areas (Falakro, Paggaio, and Menoikio). *S. scardica* was in the group of plant species that were predominant in both North-Central and Eastern areas. However, its presence was very low in Komnina and Olympos in the areas with higher collection pressure. The main determinants of floristic composition in the studied sites were altitude and precipitation, while soil features did not affect it. It was the general assumption that vegetation spreading was regulated mainly by climate at regional scales [44], while soil type and biotic interactions were regulated at the site scale [45]. Similarly, Dainese et al. [46] referred that species composition in South Alpes mostly depended on temperature and precipitation.

According to the data of this survey, presence of *S. scardica* in the six study sites was higher in habitats where mostly forbs were composing the floristic profile, and lower in the areas where graminoids and shrubs were grown. This could be directly attributed to the competition phenomena between *S. scardica* and grass species, such as *Festuca ovina* and *Sesleria* sp., but also shrubs—i.e., *Juniperus* sp.—and for below (nutrients, water) and above ground (light) resources. In general, terrestrial plant species compete at individual basis interactions for nutrients and light, and one important mechanism for maintaining local coexistence is a trade-off in competitive ability for belowground and carbon via light [47–49]. However, preliminary field studies of our group revealed that although *S. scardica* plants have no strong demand for below ground resources, their development is highly depressed from competition, especially for light (unpublished data). As light availability is a dominant factor affecting seed set [50] through the photosynthesis assimilates, grasses or shrubs present in high density may also affect the seed-set success and maturation.

Although floristic composition seemed to directly affect the presence of *S. scardica*, livestock grazing could indirectly affect the species population density and communities through its effect on the relative abundance of different plant species [51,52], resulting in competition/'eutrophication' phenomena of herbaceous plants. It has been well-documented that herbivores prevent competitive exclusion through the increment of ground-level light, especially in productive systems [53,54]. Furthermore, intraspecific plant population density may substitute a significant factor that influences interactions between plant–pollinator, pollination, and plant reproductive success [55–57]. Hence, in habitats where herbivores modify plant communities (e.g., through browsing and trampling) this modification could indirectly affect the pollination efficiency and reproductive success of plant species that have escaped herbivores.

The studied areas were mostly grazed by free-ranging cattle. Cattle grazing starts at the beginning of spring, at lower altitude grasslands, where *S. scardica* does not exist. As the available forage is restricted in lowland during summer, they move to higher altitudes, where *S. scardica* is present. However, *S. scardica* is in vegetative growth and differentiates its inflorescence organs during spring and obviously receives competitiveness from the other highly dense plant species due to the absence of grazing at that time. On the other hand, grazing in subalpine Mediterranean ecosystems has been in constant decline over the last decades [58] and in some cases, such as Tetralofos in the present study, has been completely abandoned. The abandonment of grazing in these ecosystems results in shrub encroachment, the prevalence of competing species, and the reduction of diversity [58,59]. It should be noticed that this was not the case in Tetralofos, where high floristic diversity was recorded. However, the abandonment of grazing in Tetralofos is very recent and, also, there are references that the present floristic composition indicates the historical rather than the present land use [60,61]. Nevertheless, it seems that the abandonment of grazing in these ecosystems and its impact in floristic composition could be a threat for *S. scardica* populations.

Furthermore, it was observed that in the two least accessible study areas (Falakro and Menoikio), no anthropogenic management (collections) was reported and a higher presence of *S. scardica* was

recorded. Such an observation was expected as, recently, an increment in illegal national collections of *S. scardica* was reported, which, together with false collection practices—i.e., harvesting all the plants via destructing its roots from the soil—resulted in severe fragmented populations and the unsustainable management of their natural habitats.

The phenotypic evaluation of *S. scardica*'s populations from the six case study sites revealed a high morphological diversity of leaf and inflorescence traits, as observed from the coefficient of variations (higher than 15%). Similar to our data, Aneva and Zhelev (2019) [62] reported high phenotypic diversity (coefficient of variation (CV) higher than 20%) among six wild populations of *S. scardica* in Bulgaria. Several studies report the phytochemical profiling and bioactivities of *Sideritis* sp. herbal preparations, although there are no extensive studies on the phenotypic variation and/or plasticity phenomena of the species in correlation to the micro-climate conditions. However, such phenotypic variations seem to be common also in other Lamiaceae species—for instance, *Teucrium orduini*, *Phlomis olivieri*, and *Lallemautia royleana* [63–65]—and may also arise from the genetic and/or epigenetic diversity [20,66].

## 5. Conclusions

It could be concluded that the anthropogenic management factor, through the extensive collections of aerial parts, plays a pivotal role to the declined *S. scardica* populations. In addition, this study revealed the importance of generating a better knowledge on the natural habitats of endemic vulnerable species, such as *S. scardica*, as a variable floristic composition, and its plant community's structure directly affects the presence of *S. scardica* by depressing/favoring it. The interaction of *S. scardica* with the other functional groups of plant species identified within this study indicates that the species is not competitive, and its presence can be highly depressed from the presence of graminoids, especially in high density. The abandonment of grazing in some ecosystems has a direct impact on the floristic composition, while the absence of herbivores may indirectly affect the populations of *S. scardica* through favoring the eutrophication phenomena of competitive species in natural habitats. The high phenotypic variation of *S. scardica* populations observed in this survey revealed its putative evolutionary capacity to adapt to different environments in order to maintain their survival even under unfavored conditions. As a result of the aforementioned, the present study set the basic knowledge for the development of putative in—but also *ex—situ* conservation strategies for the sustainable management of the endemic species. Although this study focuses on a detailed description of the natural habitats that *S. scardica* is endemic, further analysis on the intra- and inter-population genetic and epigenetic diversity is performed by our group, combined with a phytochemical analysis of the six populations and a metagenomic analysis of soil samples of the case study sites. Further biological experiments on the evaluation of *S. scardica*'s plasticity, but also the interactions and competition of the species with other plant commodities (i.e., grasses), would be of great importance.

**Supplementary Materials:** The following are available online at http://www.mdpi.com/2071-1050/12/6/2542/s1.

**Author Contributions:** P.K.P. was involved in the plant species identification alongside the evaluation of floristic diversity, statistical analysis of the respective data, and drafting the manuscript. E.S. was involved in the phenotypic evaluation of the populations together with statistical analysis and interpretation of the data through PCAs, drafting, and editing the manuscript. E.A. contributed to the phenotypic evaluation and drafting the manuscript. E.M.A. was involved in the conceptualization, supervision of the present research, interpretation of the data, writing, and editing the manuscript as well. All authors have read and agreed to the published version of the manuscript.

**Funding:** This research was co-financed by Greece and the European Union (European Social Fund-ESF) through the Operational Program «Human Resources Development, Education and Lifelong Learning 2014–2020» in the context of the project "Effect of environmental and management factors on the phenotypic and genetic diversity of *Sideritis scardica* Griseb. populations in North Greece" (MIS 5004926).

**Acknowledgments:** We are thankful to Ioannis Ganopoulos and Ioanna Karamichali for valuable comments on Ordination analysis and the head of Forest Services of Drama (Konstadinidou Elisavet), Serres (Giavos Konstantinos), Elassona (Ftika Zoi), and Vermio (Spiridonidis Nikolaos) for their contribution to the visit-missions in the mountainous study sites.

**Conflicts of Interest:** The authors declare no conflict of interest.

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
