# Peer review of "Abundance and Phenotypic Diversity of the Medicinal Sideritis Scardica Griseb. in Relation to Floristic Composition of Its Habitat in Northern Greece"

_sustainability, doi:10.3390/su12062542_

Round 1
Reviewer 1 Report
I have completed the review of this study. It is an interesting research regarding the abundance and phenotypic diversity of the medicinal Sideritis scardica Griseb.
The major issue that needs to be revised is the introduction of the manuscript. The introduction should be focused on what it is already known about the abundance and diversity of S. scardica instead of describing the morphological characteristics of the plant. More precisely introduction structure should include:
-a general paragraph about the abundance and diversity of the plant
-the impact of environmental factors on plant's abundance and phenotypic diversity
-the impact of anthropogenic factors on plant's abundance and phenotypic diversity
-the aim of the present study.
Thr morphological characteristics (first paragraph of the introduction in its present form) should be moved to the Materials & Methods section.
It goes without saying, that the rest of the manuscript should be reviewed accordingly.
Reviewer 2 Report
This manuscript is well-written and is including strong data collection and analysis. However in my opinion the layout of tables and figures can be improve and this will enhance the Scientific Soundness of this article.
I have some minor editorial comments and specific questions as follow?
Please follow the Sustainability Journal guideline for authors throughout the manuscript including titles and subtitles and tables and figures captions.
Title: Please follow the journal format and use upper case for each highlighted word
Keywords: Please use ; t separate the keywords instead of ,
Keywords: Please delete the .
Introduction: Line 1: Sidertis or Sideritis? I believe Sideritis is correct
Page 2 of Intro: (International Union for Conservation of Nature (IUCN) red list)
Page 2 end of paragraph 1: May you please add a reference at the end of this statement?
2.1. Study Area
Suggested caption for Figure 1:
Figure 1. The green markers are showing the location of six case study areas in North Greece.
Page 4: Please make sure how to present the chemical materials formula according to the journal requirements and be consistence K1Cr2O7 or K2Cr2O7
Please clarify how many soil samples collectively you had from different sites?
Please add (the company and the state and country) for double bean spec
Page 5: According to the literature may you determine approximately which one of these indices are more accurate for the endangered plant species?
Figure 2. Figure 3. Table4 Should be bold
Providing formula of each index in a table in supplementary materials would be helpful for readers.
Suggestion:
Supplementary Materials:Table A1 should be Table S1.

Reviewer 3 Report
Type of paper is COMMUNICATION
This communication paper contains very valuable and sophisticated work. This kind of data required a lot of effort and specific skills to choose suitable statistics software. Please double-check the paper format and style. English is fine but the style needs to be fixed.
Beside some comments such as not using Line numbers, The manuscript arrangement started to be confusing from Table6
The Heading 3- Results should be moved to the following page with the corresponding text.
Overall the main problem is the manuscript style and formatting
Round 2
Reviewer 2 Report
Authors addressed all of my concerns properly and I think the current manuscript is ready for publishing after a minor editorial corrections:
Line 52: [11,12]
Line 74: [17,18]
Line 76: [16,19]
Line 91: diversity [21].
Line 110: [23,25]
Line 127: [27,28]
Please edit the he map border.
Line 139: Figure 1.
Line 155: [33,34]
Line 176: [36,37]
Line 179: [38,39]
Line 181: [21,20]
Line 182: [38,39]
Line 185: [42,43]
Line 395. [52,53]
Line 398: [54,55]
Line 415: [61,62]
Line 614 and Line 644: Please edit this "............."section.
